# Sociodemographic differences in self-reported exposure to high fat, salt and sugar food and drink advertising: a cross-sectional analysis of 2019 UK panel data

Amy Yau ,[1,2] Jean Adams ,[3] Emma J Boyland,[4] Thomas Burgoine,[3] Laura Cornelsen ,[1] Frank de Vocht,[5,6] Matt Egan,[7] Vanessa Er,[1] Amelia A Lake,[8,9] Karen Lock,[2] Oliver Mytton,[3] Mark Petticrew,[7] Claire Thompson,[10] Martin White,[3] Steven Cummins[1]

► Prepublication history and additional supplemental materials for this paper is available online. To view these files, please visit the journal online (http://dx.doi.org/10.1136/bmjopen-2020-048139).

For numbered affiliations see end of article.

**Correspondence to**
Dr Amy Yau;
amy.yau@lshtm.ac.uk

## ABSTRACT

**Objectives** To explore sociodemographic differences in exposure to advertising for foods and drinks high in fat, salt and sugar (HFSS) and whether exposure is associated with body mass index (BMI).

**Design** Cross-sectional survey.

**Setting** UK.

**Participants** 1552 adults recruited to the Kantar Fast Moving Consumer Goods panel for London and the North of England.

**Outcome measures** Self-reported advertising exposure stratified by product/service advertised (processed HFSS foods; sugary drinks; sugary cereals; sweet snacks; fast food or digital food delivery services) and advertising setting (traditional; digital; recreational; functional or transport); BMI and sociodemographic characteristics.

**Results** Overall, 84.7% of participants reported exposure to HFSS advertising in the past 7 days. Participants in the middle (vs high) socioeconomic group had higher odds of overall self-reported exposure (OR 1.48; 95% CI 1.06 to 2.07). Participants in the low (vs high) socioeconomic group had higher odds of reporting exposure to advertising for three of five product categories (ORs ranging from 1.41 to 1.67), advertising for digital food delivery services (OR 1.47; 95% CI 1.05 to 2.05), traditional advertising (OR 1.44; 95% CI 1.00 to 2.08) and digital advertising (OR 1.50; 95% CI 1.06 to 2.14). Younger adults (18–34 years vs ≥65 years) had higher odds of reporting exposure to advertising for digital food delivery services (OR 2.08; 95% CI 1.20 to 3.59), digital advertising (OR 3.93; 95% CI 2.18 to 7.08) and advertising across transport networks (OR 1.96; 95% CI 1.11 to 3.48). Exposure to advertising for digital food delivery services (OR 1.40; 95% CI 1.05 to 1.88), digital advertising (OR 1.80; 95% CI 1.33 to 2.44) and advertising in recreational environments (OR 1.46; 95% CI 1.02 to 2.09) was associated with increased odds of obesity.

**Conclusions** Exposure to less healthy product advertising was prevalent, with adults in lower socioeconomic groups and younger adults more likely to report exposure. Broader restrictions may be needed to

### Strengths and limitations of this study

► This study investigates exposure to advertising for a range of less healthy products in various settings, including some that have been less studied (eg, advertising for digital food delivery services and advertising across transport networks).

► Self-reported exposure may not be an accurate reflection of actual exposure to advertising due to poor recall or social desirability bias.

► The cross-sectional design of this study limits our interpretation of the findings as reverse causality may explain the associations we observed.

► Participants were from two regions (London and the North of England), so may not be representative of the UK as a whole.

reduce sociodemographic differences in exposure to less healthy product advertising.

## INTRODUCTION

In recent years there has been an increased focus on the potential contribution of marketing and advertising of less healthy foods and drinks, such as those high in fat, salt and sugar (HFSS), to population diet and diet-related disease.[1 2] Over £300 million was spent on advertising of less healthy foods and drinks in the UK in 2017.[3 4] This spend substantially outweighs that for healthier products. For example, £12 million was spent on advertising vegetables compared with almost £87 million on soft drinks in the UK in 2015.[4] Advertising and marketing of less healthy foods and drinks to children is widespread and is associated with children's preferences, requests for purchases and consumption.[5–8] In adults, evidence is

mixed, but some studies have found that exposure to less healthy food and drink advertising influences purchasing and consumption, and normalises the consumption of less healthy foods and drinks.[9–14] Advertising exposure may therefore be a plausible influence on obesity and diet-related non-communicable diseases.[15 16] A systematic review found that children from ethnic minority and socioeconomically disadvantaged backgrounds were disproportionately exposed to advertising for less healthy foods.[17] In adults, exposure to outdoor advertising and television advertising of less healthy foods is correlated with socioeconomic position in the UK.[18 19] These findings suggest that differential exposure to less healthy advertising may be contributing to the higher burden of obesity and diet-related non-communicable diseases among lower socioeconomic groups.[20–22]

In response, the regulation of food and drink marketing and advertising has been increasingly promoted as a policy lever for obesity prevention by local and national governments and the WHO.[8 23 24] In the UK, policies have focused on highly processed HFSS products as determined by the Nutrient Profiling Model.[23 25] However, restrictions on other advertising may also be needed. The use and advertising of digital food delivery services have increased dramatically in recent years.[26] These services often deliver takeaway fast-food or restaurant meals, which tend to be less healthy than home-prepared food.[27–29] Therefore, regulating the promotion of digital food delivery services may also be a potential policy lever to reduce consumption of less healthy products.

A wide range of advertising channels are used by companies to promote their products and brand. The use of multiple advertising settings both reinforces the messaging and helps reach a wider audience.[30] Outdoor advertising is thought to reach 98% of the UK population at least once a week and is especially effective at reaching young, urban, affluent consumers.[31] Advertising across the Transport for London network generated £152.1 million in 2017/2018, accounting for 40% of London's outdoor advertising spend and 20% of outdoor advertising spend across the UK.[32] Digital advertising has grown, with online advertisers spending £13.4 billion in 2018 in the UK, and is set to grow further.[33 34] Digital advertising has the advantage of allowing more customisation and is often targeted towards younger people, especially those in more disadvantaged areas.[35–37] Advertising through more traditional media, such as television, remains popular with food companies and is considered cost-effective and wide reaching.[33]

Obesity and diet quality are socially patterned, with greater prevalence of obesity and poorer diet quality associated with lower socioeconomic position.[21 38] Socioeconomic inequalities in exposure to advertising of less healthy foods and drinks may be a mechanism for the generation of socioeconomic inequalities in diet and obesity. While there is some evidence that levels of exposure to outdoor advertising and television advertising for HFSS products differ by socioeconomic position, the evidence base for sociodemographic correlates of advertising exposure in the UK remains limited.[18 19] Very little research has been conducted to explore differences in exposure to a wider range of advertising settings, such as digital advertising and advertising across transport networks.[39] Exposure to advertising for digital food delivery services is also poorly understood. Further, few studies have examined sociodemographic differences in advertising exposure beyond socioeconomic position. There has also been limited work that directly explores associations between exposure to less healthy food and drink advertising and obesity. In this paper, we address these gaps by using survey data from the UK (London and the North of England) to explore sociodemographic differences in: (1) overall exposure to less healthy food and drink advertising, (2) advertising exposure stratified by product category, (3) exposure to advertising for digital food delivery services, and (4) advertising exposure stratified by advertising setting. We then examine the association between advertising exposure and body mass index (BMI).

## METHODS
### Data
We used baseline cross-sectional data from a study evaluating the impact of restricting advertising of HFSS foods and drinks across the Transport for London network.[32 40] Data were from households recruited to the UK Kantar Fast Moving Consumer Goods panel for London and the North of England (n=1552 households). Our sample size was based on the maximum number of available households within the panel for the two regions. The main food shopper from each recruited household was asked to complete an online survey (between 10 and 18 February 2019) on recent exposure to advertising of less healthy foods and drinks. The response rate was 71%. Households were recruited to the consumer panel through stratified random sampling and were representative of the regions from which they were sampled on the basis of household size, number of children, socioeconomic position and age of main shopper.[41 42] Panel households are recruited by Kantar through post and email.[42]

### Sociodemographic characteristics
Based on self-reported survey data, participants were categorised by sociodemographic characteristics: sex (male and female), age group (18–34, 35–44, 45–54, 55–64 and ≥65 years), children <16 years in the household (yes or no), socioeconomic position and working status. For socioeconomic position, participants were classified according to the National Readership Survey (NRS) occupational social grade classification (A, B, C1, C2, D, E).[43] We categorised NRS social grade into three socioeconomic groups: high (AB), middle (C1C2) and low (DE). For working status, we categorised participants into six categories: full-time employee, part-time employee, self-employed, retired, not looking for work or

unable to work (looking after home or family, long-term sick or disabled, away from work due to illness, maternity leave, holiday or unemployed and not looking for work) or other (government-sponsored training scheme, other paid work, student, actively looking for paid work or other).

### Region

From household postcode data, participants were categorised as living in either London (Greater London) or the North of England (North West, North East, or Yorkshire and the Humber).

### Measuring exposure to advertising

We assessed exposure to advertising for five of the most commonly advertised food and drink product categories of current policy concern: processed HFSS foods, sugary drinks, sugary cereals, sweet snacks and fast food.[44] Definitions of the product categories used in the survey were adapted from the International Food Policy Study and are available in online supplemental table S1.[45] In addition, exposure to advertising for digital food delivery services was assessed with a question on exposure to advertising for food delivery apps, with the market-leading services (Uber Eats, Deliveroo, Just Eat and Foodhub) listed as examples.

Survey questions were structured as follows: '*In the last 7 days, have you seen or heard advertisements for* [category] *in the following places?*' Within each question, 19 places where advertisements may have been seen or heard were specified. Using methods adapted from Forde *et al*,[12] we recategorised the 19 places into five advertising settings prior to analysis: traditional advertising, digital advertising, advertising in recreational environments, advertising in functional environments and advertising across transport networks (table 1). Participants were also able to report other places using free text. We allocated these text responses into the five advertising settings, treating the categories as mutually exclusive. Where there was

ambiguity in categorisation, advertising setting was categorised based on *where* the advertisement was seen or heard rather than what medium was used. For example, a digital advertisement *at a bus stop* was categorised as transport.

We coded advertising exposure into two categories: exposed or not. For overall exposure, participants who reported seeing or hearing advertisements for any of the six product/service categories (processed HFSS foods; sugary drinks; sugary cereals; sweet snacks; fast-food and digital food delivery services) were classified as exposed.

### BMI and weight status

BMI (weight/height$^2$) was calculated using self-reported height and weight data. BMI was available for 81.7% of participants (n=1268), who were then classified into four categories: underweight (BMI <18.5), normal (BMI ≥18.5 and <25), overweight (BMI ≥25 and <30) and obese (BMI ≥30). A fifth category contained participants with missing BMI data (n=284).

### Statistical methods

We calculated the number (%) of participants reporting exposure to less healthy food and drink advertising (including digital food delivery services) in the past 7 days, overall and stratified by product category and advertising setting. Using logistic regression models we estimated ORs, with 95% CIs, for the association between sociodemographic characteristics and advertising exposure. As advertising exposure may be influenced by social and environmental factors, independently or in combination, we tested for interactions between sociodemographic characteristics and region (London or the North of England). Separate models were used to look at overall advertising exposure, advertising exposure stratified by product category, exposure to advertising for digital food delivery services and exposure stratified by advertising setting. We also used logistic regression models to investigate the association between self-reported advertising

| Table 1 | Categorisation of advertising setting | |
|---|---|---|
| **Advertising setting** | **Description** | **Included survey responses** |
| Traditional | Physical, non-digital text and radio media and direct marketing | Television, radio, text message, newspaper/magazine, email and leaflet |
| Digital | Advertising seen or heard through digital platforms and social media | Online/internet, mobile app, video game and social media |
| Recreational | Advertising placed in environments that people interact with for enjoyment and leisure purposes | Film/cinema, leisure centre/gym/community centre, sports event/concert/community event, giveaway/sample/special offer and pub |
| Functional | Advertising placed in environments that people visit for a specific purpose and to complete a specific task (eg, school, work and shops) | Billboard/outdoor signs, telephone boxes, school/college/university, signs or displays in supermarket/convenience stores/restaurants, delivery drivers, doctor's surgery, shopping centre and motorway services |
| Transport | Advertising placed in environments related to transport | Outside/inside buses, outside/inside tube, tram or train, outside/inside of tube or train station, bus stop, taxi and back of bus ticket |

**Table 2** Sociodemographic characteristics of study population (n=1552)

| Sociodemographic characteristic | | Total, n (%) | London, n (%) | North, n (%) | $X^2$ (p value) |
|---|---|---|---|---|---|
| Sex | Male | 441 (28.4) | 213 (30.2) | 228 (26.9) | 2.05 (0.152) |
| | Female | 1111 (71.6) | 492 (69.8) | 619 (73.1) | |
| Age group (years) | 18–34 | 188 (12.1) | 65 (9.20) | 123 (14.5) | 18.73 (0.001)*** |
| | 35–44 | 299 (19.3) | 160 (22.7) | 139 (16.4) | |
| | 45–54 | 411 (26.5) | 196 (27.8) | 215 (25.4) | |
| | 55–64 | 335 (21.6) | 145 (20.6) | 190 (22.4) | |
| | ≥65 | 319 (20.6) | 139 (19.7) | 180 (21.3) | |
| Socioeconomic position | AB | 341 (22.0) | 177 (25.1) | 164 (19.4) | 7.92 (0.019)* |
| | C1C2 | 926 (59.7) | 409 (58.0) | 517 (61.0) | |
| | DE | 285 (18.4) | 119 (16.9) | 166 (19.6) | |
| Children in the household | No | 1110 (71.5) | 507 (71.9) | 603 (71.2) | 0.10 (0.754) |
| | Yes | 442 (28.5) | 198 (28.1) | 244 (28.8) | |
| Working status | Full time | 612 (39.4) | 285 (40.4) | 327 (36.8) | 19.88 (0.001)*** |
| | Part time | 223 (14.4) | 91 (12.9) | 132 (15.6) | |
| | Self-employed | 131 (8.4) | 78 (11.6) | 53 (6.3) | |
| | Retired | 342 (22.0) | 145 (20.6) | 197 (23.3) | |
| | Not looking/unable to work | 214 (13.8) | 87 (12.3) | 127 (15.0) | |
| | Other | 30 (1.9) | 19 (2.7) | 11 (1.3) | |
| BMI | Underweight | 31 (2.0) | 21 (3.0) | 10 (1.2) | 11.91 (0.018)* |
| | Normal | 479 (30.9) | 236 (33.5) | 243 (28.7) | |
| | Overweight | 425 (27.4) | 180 (25.5) | 245 (28.9) | |
| | Obese | 336 (21.7) | 143 (20.3) | 193 (22.8) | |
| | Missing | 281 (18.1) | 125 (17.7) | 156 (18.4) | |
| Region | North | 847 (54.6) | N/A | N/A | N/A |
| | London | 705 (45.4) | | | |

*P<0.05, ** P<0.01, ***P<0.001.
BMI, body mass index; N/A, not applicable.

exposure and odds of living with overweight or obesity. All regression models were mutually adjusted for sex, age group, socioeconomic position, children in the household, working status and region. All analyses were conducted in Stata IC V.16 and completed in September 2020.

## Patient and public involvement
Patients and the public were not involved in the design, or conduct, or reporting, or dissemination plans of our research.

## RESULTS
Overall, 1552 participants were included in this study (table 2) with 45.4% living in London and 54.6% in the North of England. The majority of participants were female (71.6%), in the middle socioeconomic group (C1C2) (59.7%), had no children <16 years in the household (71.5%) and were in work (62.2%).

## Self-reported exposure to advertising
Overall, 84.7% of participants reported seeing or hearing advertising for less healthy foods and drinks and/or digital food delivery services in the past 7 days (see online supplemental table S2). The proportion of participants reporting exposure to advertising differed according to product category: 68.2% for processed HFSS foods, 52.4% for sugary drinks, 42.1% for sugary cereals, 55.0% for sweet snacks, 71.3% for fast food and 54.9% for digital food delivery services. Reported exposure also varied according to advertising setting: 74.0% of participants reported seeing or hearing traditional advertising, 38.7% digital advertising, 18.8% advertising in recreational environments, 51.5% advertising in functional environments and 36.4% advertising across transport networks (see online supplemental table S3). For most product categories, reported exposure was most common through traditional advertising, followed by advertising in functional environments, digital advertising, advertising across transport networks and, lastly, advertising in recreational

environments. However, for digital food delivery services, digital was the second highest reported setting after traditional. Self-reported exposure to advertising was higher in London than the North of England for sugary drinks (55.3% vs 49.9%, p=0.035), digital food delivery services (59.0% vs 51.5%, p=0.003) and advertising across transport networks (45.5% vs 28.8%, p<0.001) (see online supplemental table S4). After adjustments, participants living in London had higher odds of self-reported exposure for sugary drinks (OR 1.27; 95% CI 1.03 to 1.56), digital food delivery services (OR 1.39; 95% CI 1.13 to 1.71) and advertising across transport networks (OR 2.05; 95% CI 1.65 to 2.54) compared with those living in the North of England (tables 3 and 4).

### Sociodemographic differences in self-reported exposure to advertising for less healthy foods and drinks

The adjusted odds of self-reported exposure to advertising overall were higher in the middle socioeconomic group (C1C2) compared with the high socioeconomic group (AB) (OR 1.48; 95% CI 1.06 to 2.07) (table 3). When stratified by product category, sociodemographic differences in self-reported exposure to advertising were found in three of the five categories studied (processed HFSS foods, sugary cereals and sweet snacks). Socioeconomic differences were observed for these three product categories, with lower socioeconomic position associated with higher adjusted odds of self-reported exposure (ORs ranging from 1.41 to 1.67 for low compared with high). There was indication of a socioeconomic gradient for all three product categories, although the CIs for the middle and low socioeconomic groups overlapped. Gender differences were also observed for processed HFSS foods, with higher adjusted odds for women compared with men (OR 1.44; 95% CI 1.13 to 1.84). For sweet snacks, participants who were full-time employees had marginally higher adjusted odds of reporting advertising exposure compared with participants who were not looking for work or unable to work (OR 1.40; 95% CI 1.00 to 1.97). There were no observed associations between sociodemographic characteristics and self-reported exposure to advertising for fast-food or sugary drinks. There were some interactions between sociodemographic characteristics and region on their influence on self-report advertising exposure (see online supplemental table S5). Results stratified by region are presented in online supplemental tables S6 and S7 where significant.

### Sociodemographic differences in exposure to advertising for digital food delivery services

Higher adjusted odds of reporting exposure to advertising for digital food delivery services were observed in the lower socioeconomic groups compared with the highest: low (OR 1.47; 95% CI 1.05 to 2.05) and middle (OR 1.39; 95% CI 1.08 to 1.80). The adjusted odds were also higher among participants aged 18–34 years (OR 2.08; 95% CI 1.20 to 3.59), 35–44 years (OR 1.93; 95% CI

1.15 to 3.36) and 55–64 years (OR 1.53; 95% CI 1.00 to 2.35) compared with those aged ≥65 years.

### Sociodemographic differences in exposure to advertising by advertising setting

Sociodemographic differences in self-reported exposure were found for traditional advertising, digital advertising and advertising across transport networks (table 4). Adjusted odds of self-reported exposure were higher in the lower socioeconomic groups compared with the high socioeconomic group for traditional advertising: low (OR 1.44; 95% CI 1.00 to 2.08) and middle (OR 1.52; 95% CI 1.15 to 2.00). Adjusted odds of self-reported exposure to digital advertising were higher in the low compared with high socioeconomic group (OR 1.50; 95% CI 1.06 to 2.14), and younger age groups compared with ≥65 years: 18–34 years (OR 3.93; 95% CI 2.18 to 2.07) and 35–44 years (OR 3.06; 95% CI 1.74 to 5.40). The adjusted odds of reporting exposure to advertising across transport networks were higher among participants aged 18–34 years compared with ≥65 years (OR 1.96; 95% CI 1.11 to 3.48), and participants who were full-time employees compared with those not looking for work or unable to work (OR 1.50; 95% CI 1.04 to 2.17).

### Association between advertising exposure and BMI

Overall self-reported exposure to less healthy food and drink advertising was not associated with BMI category (table 5). However, higher adjusted odds of living with obesity were observed among participants who reported exposure to advertising for digital food delivery services (OR 1.40; 95% CI 1.05 to 1.88), digital advertising (OR 1.80; 95% CI 1.33 to 2.44) and advertising in recreational environments (OR 1.46; 95% CI 1.02 to 2.09).

## DISCUSSION
### Summary of findings
This is one of the few studies to investigate sociodemographic correlates of exposure to less healthy food and drink advertising in UK adults. Overall, exposure was high, with 84.7% of participants reporting seeing or hearing advertising for less healthy foods and drinks and/ or digital food delivery services in the past 7 days. Lower socioeconomic position was associated with increased odds of advertising exposure. Overall, the middle (C1C2) socioeconomic group had higher odds of exposure compared with the most affluent group (AB). For three of the five food and drink product categories (processed HFSS foods, sugary cereals and sweet snacks) and digital food delivery services, increased odds of self-reported exposure to advertising were observed for lower socioeconomic groups (DE and C1C2 compared with AB). When stratified by advertising setting, higher odds of reporting exposure among lower socioeconomic groups were observed for traditional (C1C2 and DE) and digital advertising (DE only). Younger participants had higher odds of self-reported exposure to advertising for digital

**Table 3** Sociodemographic correlates of advertising exposure stratified by product/service advertised (n=1552)

| Sociodemographic characteristic | | Any | Processed HFSS foods | Sugary drinks | Sugary cereals | Sweet snacks | Fast food | Digital food delivery services |
|---|---|---|---|---|---|---|---|---|
| Sex | Male | Ref | Ref | Ref | Ref | Ref | Ref | Ref |
| | Female | 1.11 (0.81–1.53) | 1.44 (1.13–1.84)** | 1.09 (0.86–1.37) | 0.89 (0.70–1.12) | 1.16 (0.92–1.46) | 1.02 (0.79–1.31) | 0.93 (0.74–1.18) |
| Age group | 18–34 | 1.11 (0.52–2.35) | 1.21 (0.68–2.17) | 1.07 (0.62–1.83) | 1.12 (0.65–1.94) | 0.99 (0.57–1.70) | 1.24 (0.68–2.24) | 2.08 (1.20–3.59)** |
| | 35–44 | 1.21 (0.59–2.50) | 1.39 (0.80–2.44) | 1.09 (0.65–1.83) | 1.41 (0.83–2.39) | 0.96 (0.57–1.61) | 1.29 (0.73–2.29) | 1.93 (1.15–3.26)** |
| | 45–54 | 1.09 (0.56–2.12) | 0.97 (0.58–1.61) | 0.96 (0.59–1.54) | 1.18 (0.73–1.92) | 0.86 (0.54–1.39) | 1.25 (0.74–2.12) | 1.40 (0.87–2.26) |
| | 55–64 | 1.05 (0.58–1.90) | 1.16 (0.73–1.84) | 1.21 (0.79–1.85) | 1.39 (0.90–2.14) | 1.06 (0.69–1.62) | 1.42 (0.88–2.28) | 1.53 (1.00–2.35)* |
| | ≥65 | Ref | Ref | Ref | Ref | Ref | Ref | Ref |
| Socioeconomic position | AB | Ref | Ref | Ref | Ref | Ref | Ref | Ref |
| | C1C2 | 1.48 (1.06–2.07)* | 1.42 (1.09–1.84)** | 1.22 (0.95–1.57) | 1.50 (1.16–1.95)** | 1.31 (1.02–1.68)* | 1.25 (0.95–1.64) | 1.39 (1.08–1.80)** |
| | DE | 1.30 (0.84–2.02) | 1.67 (1.17–2.38)** | 1.36 (0.97–1.88) | 1.54 (1.10–2.16)** | 1.41 (1.01–1.96)* | 1.28 (0.90–1.84) | 1.47 (1.05–2.05)* |
| Children in the household | No | Ref | Ref | Ref | Ref | Ref | Ref | Ref |
| | Yes | 0.72 (0.50–1.04) | 0.78 (0.59–1.05) | 0.82 (0.63–1.08) | 1.07 (0.81–1.40) | 0.91 (0.69–1.20) | 0.79 (0.59–1.06) | 0.77 (0.59–1.02) |
| Working status | Full time | 1.05 (0.66–1.66) | 1.13 (0.78–1.63) | 1.17 (0.81–1.59) | 1.17 (0.83–1.65) | 1.40 (1.00–1.97)* | 1.12 (0.77–1.63) | 1.22 (0.87–1.72) |
| | Part time | 1.00 (0.60–1.69) | 0.96 (0.64–1.45) | 0.97 (0.66–1.42) | 1.01 (0.69–1.50) | 1.04 (0.71–1.53) | 0.96 (0.63–1.46) | 0.93 (0.63–1.37) |
| | Self-employed | 1.13 (0.60–2.12) | 0.80 (0.50–1.29) | 0.88 (0.56–1.38) | 0.89 (0.56–1.40) | 1.09 (0.70–1.70) | 1.18 (0.72–1.95) | 0.84 (0.54–1.32) |
| | Retired | 1.06 (0.53–2.11) | 1.24 (0.72–2.13) | 1.06 (0.64–1.73) | 1.33 (0.81–2.20) | 1.19 (0.72–1.96) | 1.13 (0.65–1.96) | 1.16 (0.70–1.91) |
| | Not looking/ unable to work | Ref | Ref | Ref | Ref | Ref | Ref | Ref |
| | Other | 1.13 (0.37–3.49) | 0.80 (0.36–1.80) | 1.18 (0.54–2.57) | 1.27 (0.58–2.76) | 1.30 (0.60–2.84) | 0.70 (0.31–1.57) | 1.00 (0.45–2.18) |
| Region | London | Ref | Ref | Ref | Ref | Ref | Ref | Ref |
| | North | 1.26 (0.95–1.68) | 1.10 (0.88–1.37) | 1.27 (1.03–1.56)* | 1.05 (0.86–1.29) | 0.91 (0.74–1.12) | 1.09 (0.87–1.36) | 1.39 (1.13–1.71)** |

Models adjusted for sex, age group, social class, children in the household, working status and region.
*P<0.05; **p<0.01; ***p<0.001.
HFSS, high in fat, salt and sugar.

**Table 4** Sociodemographic correlates of advertising exposure stratified by advertising setting (n=1552)

| Sociodemographic characteristic | | Traditional | Digital | Recreational | Functional | Transport |
|---|---|---|---|---|---|---|
| Sex | Male | Ref | Ref | Ref | Ref | Ref |
| | Female | 1.19 (0.92–1.54) | 0.80 (0.63–1.02) | 0.94 (0.71–1.26) | 0.96 (0.76–1.21) | 0.89 (0.70–1.14) |
| Age group | 18–34 | 0.64 (0.34–1.18) | 3.93 (2.18–7.08)*** | 1.37 (0.70–2.70) | 1.27 (0.74–2.18) | 1.96 (1.11–3.48)* |
| | 35–44 | 0.74 (0.41–1.34) | 3.06 (1.74–5.40)*** | 1.27 (0.66–2.44) | 1.12 (0.67–1.88) | 1.50 (0.87–2.61) |
| | 45–54 | 0.77 (0.44–1.34) | 1.69 (0.99–2.87) | 0.84 (0.46–1.55) | 0.89 (0.56–1.43) | 0.99 (0.59–1.65) |
| | 55–64 | 0.94 (0.57–1.54) | 1.38 (0.85–2.24) | 1.13 (0.66–1.94) | 0.99 (0.65–1.51) | 0.94 (0.59–1.49) |
| | ≥65 | Ref | Ref | Ref | Ref | Ref |
| Socioeconomic position | AB | Ref | Ref | Ref | Ref | Ref |
| | C1C2 | 1.52 (1.15–2.00)** | 1.22 (0.93–1.60) | 1.01 (0.73–1.39) | 1.16 (0.90–1.49) | 1.08 (0.82–1.41) |
| | DE | 1.44 (1.00–2.08)* | 1.50 (1.06–2.14)* | 1.04 (0.68–1.59) | 1.18 (0.85–1.65) | 1.12 (0.78–1.59) |
| Children in the household | No | Ref | Ref | Ref | Ref | Ref |
| | Yes | 0.93 (0.69–1.26) | 0.96 (0.72–1.27) | 0.74 (0.52–1.05) | 0.88 (0.67–1.15) | 0.79 (0.60–1.06) |
| Working status | Full time | 1.01 (0.68–1.48) | 1.28 (0.90–1.82) | 1.18 (0.76–1.82) | 1.07 (0.76–1.49) | 1.50 (1.04–2.17)* |
| | Part time | 0.85 (0.55–1.31) | 1.08 (0.73–1.61) | 1.05 (0.64–1.73) | 0.84 (0.57–1.23) | 1.12 (0.74–1.71) |
| | Self-employed | 0.80 (0.49–1.33) | 0.93 (0.58–1.48) | 0.87 (0.48–1.58) | 0.95 (0.61–1.48) | 1.55 (0.97–2.49) |
| | Retired | 0.90 (0.50–1.60) | 0.84 (0.49–1.45) | 0.93 (0.49–1.76) | 0.80 (0.49–1.32) | 1.02 (0.59–1.76) |
| | Not looking/unable to work | Ref | Ref | Ref | Ref | Ref |
| | Other | 0.76 (0.32–1.78) | 0.58 (0.25–1.36) | 1.61 (0.66–3.94) | 0.85 (0.39–1.84) | 2.18 (0.98–4.86) |
| Region | London | Ref | Ref | Ref | Ref | Ref |
| | North | 1.13 (0.90–1.43) | 1.20 (0.97–1.49) | 1.11 (0.86–1.45) | 1.16 (0.94–1.42) | 2.05 (1.65–2.54)*** |

Models adjusted for sex, age group, social class, children in the household, working status and region.
*P<0.05; **p<0.01; ***p<0.001.

**Table 5** Weight status by exposure to advertising for less healthy foods and drinks (n=1552)

| Type of advertising | Underweight, OR (95% CI) | Overweight, OR (95% CI) | Obese, OR (95% CI) | Missing, OR (95% CI) |
|---|---|---|---|---|
| Any advertising exposure | 0.94 (0.35 to 2.57) | 1.08 (0.75 to 1.57) | 1.08 (0.72 to 1.60) | 0.94 (0.63 to 1.41) |
| By product category | | | | |
| Processed HFSS foods | 1.31 (0.56 to 3.04) | 1.17 (0.87 to 1.56) | 1.12 (0.82 to 1.53) | 0.89 (0.65 to 1.21) |
| Sugary drinks | 1.07 (0.51 to 2.24) | 1.09 (0.84 to 1.43) | 1.00 (0.75 to 1.33) | 1.04 (0.78 to 1.41) |
| Sugary cereals | 1.03 (0.49 to 2.20) | 1.21 (0.92 to 1.59) | 1.29 (0.97 to 1.72) | 1.03 (0.76 to 1.40) |
| Sweet snacks | 1.14 (0.54 to 2.39) | 1.03 (0.79 to 1.35) | 1.11 (0.83 to 1.47) | 0.79 (0.58 to 1.06) |
| Fast food | 0.69 (0.32 to 1.47) | 1.18 (0.88 to 1.58) | 1.31 (0.95 to 1.80) | 1.00 (0.72 to 1.38) |
| Digital food delivery services | 0.71 (0.33 to 1.49) | 1.22 (0.93 to 1.60) | 1.40 (1.05 to 1.88)* | 1.05 (0.78 to 1.43) |
| By advertising setting | | | | |
| Traditional | 0.88 (0.40 to 1.94) | 1.32 (0.98 to 1.79) | 1.28 (0.93 to 1.77) | 1.24 (0.89 to 1.74) |
| Digital | 1.12 (0.51 to 2.47) | 1.28 (0.96 to 1.71) | 1.80 (1.33 to 2.44)*** | 1.12 (0.82 to 1.54) |
| Recreational | 1.05 (0.41 to 2.66) | 1.15 (0.81 to 1.63) | 1.46 (1.02 to 2.09)* | 1.07 (0.72 to 1.58) |
| Functional | 0.65 (0.31 to 1.38) | 1.01 (0.78 to 1.32) | 1.15 (0.86 to 1.52) | 1.06 (0.79 to 1.43) |
| Transport | 0.63 (0.28 to 1.41) | 1.07 (0.80 to 1.42) | 1.20 (0.89 to 1.63) | 0.78 (0.57 to 1.08) |

Model adjusted for sex, age group, social class, children in the household, working status and region.
*P<0.05; **p<0.01; ***p<0.001.
HFSS, high in fat, salt and sugar.

food delivery services compared with participants aged ≥65 years, except those aged 45–54 years. Younger participants were also more likely to report exposure to digital advertising (18–34 and 35–44 age groups) and advertising across transport networks (18–34 age group). Participants who were full-time employees had higher self-reported odds of exposure to advertising for sweet snacks and advertising across transport networks. Women had higher odds of reporting exposure to advertising for processed HFSS foods than men. Exposure to advertising for digital food delivery services, digital advertising and advertising in recreational environments was associated with obesity.

### Strengths and limitations

While most of the existing research on less healthy food and drink advertising focuses on children, this study adds to the evidence for sociodemographic differences in advertising exposure in adults. There are some strengths and limitations of this study that should be noted. There is no standard method for measuring advertising exposure. The questions used in this study were informed by those used in the International Food Policy Study.[12] The survey questions asked about the products advertised and the advertising setting, giving us the opportunity to explore what was advertised and where. This provided a broader perspective relative to studies focusing on one form of advertising.[18 19] The inclusion of digital food delivery services in the survey also provided insight into a fast-growing and understudied channel for purchasing less healthy foods and drinks.

The cross-sectional nature of this study and short recall period may mean that exposure as measured is not representative of typical exposure, for example, due to the seasonality of advertising.[46] Furthermore, we did not account for the intensity of exposure, as we only categorised participants as exposed or not, and did not measure how many advertisements participants saw or heard. The cross-sectional nature of this study also limits our ability to establish causality and reverse causation could explain some of the associations observed.

Self-reported information is subject to misreporting. For example, height is often over-reported and weight under-reported, leading to inaccuracies in BMI.[47] Therefore, our interpretation of the associations found in this study should be cautious. We also only used one proxy measure of socioeconomic position and therefore may not have fully captured socioeconomic variation in advertising exposure. Further studies using objectively measured BMI and advertising exposure, and other proxy measures of socioeconomic position such as educational level, should be conducted to confirm our findings. Other sociodemographic characteristics not available in our study, such as ethnicity, may also be associated with advertising exposure and warrant study in future.

### Generalisability

Our study sample was likely representative of populations in London and the North of England, as Kantar assesses representativeness of their panels every 4 weeks.[42] However, our study sample was not representative of the UK as a whole. Nonetheless, our study sample appears similar to national samples and the general population in terms of distribution of most sociodemographic characteristics. Most participants (71.6%) were female, which is likely due to more women being the main food shopper within a household than men. A similar proportion of

main food providers were women (73.3%) in latest wave (2014/2015 to 2016/2017) of the National Diet and Nutrition Survey (NDNS), which aims to be representative of the UK population.[48] A similar proportion of participants also reported not having children in the household in our sample (71.5%) as in the NDNS (69.8%), but this is higher than in the general population (58.0%).[49] Having children may impact on advertising exposure (eg, through different advertisements shown around children's television programmes) or recall of advertising exposure (eg, through 'pester power').[50 51] A greater proportion of participants in our sample reported being in work (62.2%) than in the NDNS (56.2%). Discounting those with missing BMIs, the proportion of participants in each BMI category in this study was similar to that found in the Health Survey for England 2018.[21] However, the number of participants in the underweight category was too small for any statistically meaningful interpretations to be drawn for this group in our study.

### Interpretation

Socioeconomic differences in self-reported advertising exposure were observed across various product categories and advertising settings, with more disadvantaged groups more likely to report exposure than the least disadvantaged group. These findings are consistent with previous studies that documented socioeconomic differences in exposure to less healthy food and drink advertising in the UK using more objective measures of exposure.[18 19] The alignment between objective and self-reported measures suggests that self-reported measures may be appropriate proxies for advertising exposure. Further, our findings

are consistent with those in the wider international literature.[17 52 53]

There are various possible explanations for the associations between sociodemographic characteristics and self-reported advertising exposure observed in this study (figure 1). Low socioeconomic position, younger age, full-time working and being female may be driving higher exposure to less healthy advertising. For example, for socioeconomic position, this could be through a greater concentration of advertising for less healthy products in more disadvantaged areas. Studies in the USA have found that advertising for HFSS foods and drinks was more prevalent in more deprived neighbourhoods,[19] and that the types of products advertised also differ depending on neighbourhood demographics.[54] We did not capture the urbanicity of participants' locations in this study, which may also influence the density and types of outdoor advertising present in the local neighbourhood.[54] Another potential influence on actual advertising exposure is differential use of public transport across population subgroups. Participants who were full-time employees were more likely to report exposure to advertising across transport networks, as were younger participants. These groups may have been more exposed to advertising across transport networks due to more frequent public transport use.[55 56] Differences in advertising exposure could also be associated with known sociodemographic differences in screen time and use of certain media, such as radio and social media.[57 58]

Alternatively, the observed differences may be due to differences in recall rather than actual exposure. The

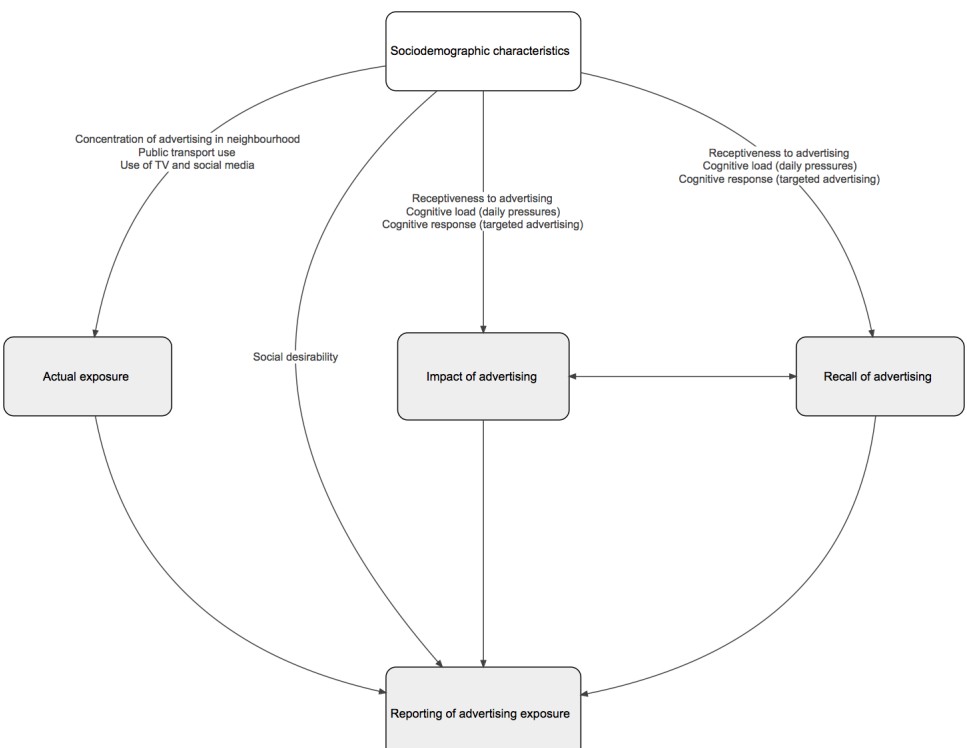

**Figure 1**  Potential explanations for sociodemographic differences in exposure to advertising for less healthy foods and drinks.

content of advertisements may be designed to appeal to certain population groups resulting in different cognitive responses to advertisements. Identity salience (defined as heightened sensitivity to identity-relevant stimuli) may make advertisements more impactful to certain groups (ie, advertisements that feel relevant may be more impactful) and therefore more memorable.[59] Some groups may also be more receptive to advertising due to higher cognitive load. One experiment showed greater effect of advertising on food choices when cognitive load was higher.[11] The authors argued that the effect of advertising on food choices could therefore be exacerbated in low socioeconomic groups because of the greater cognitive load of daily pressures. Receptiveness to advertising may also decrease with age.[36] Equally, sociodemographic differences in reporting may reflect differential social desirability bias across population groups.

Differences in reported exposure to advertising across transport networks between London and the North of England may be partly explained by differences in the extent of public transport infrastructure, with London having the most extensive transport network in the UK.[32] Participants living in London also reported greater exposure to advertising for digital food delivery services. This may be due to the greater penetration of digital food delivery services in London while these services are continuing to expand.[60]

Self-reported exposure to advertising for digital food delivery services, digital advertising and advertising in recreational environments was independently associated with higher odds of obesity. While it is implausible that advertising exposure over 7 days directly caused obesity, advertising exposure may be stable over time and suggestive of a long-term effect of advertising on the consumption of less healthy products. Reverse causality, whereby people living with obesity are more likely to see or hear, recall or report advertising of less healthy products, may also explain the associations observed.

A large majority of participants reported seeing or hearing some form of advertising for less healthy foods and drinks and/or digital food delivery services, indicating high prevalence of exposure across the UK adult population. This level of exposure was consistent with levels found elsewhere using self-reported measures.[12 61] However, advertising can influence consumer behaviour unconsciously.[62] Therefore, self-reported measures may underestimate advertising exposure. When objectively measured, one study found that children were exposed to HFSS advertising 27.3 times per day.[63]

The variety of advertising strategies used by companies to advertise HFSS products suggests that restrictions may need to be broadened beyond those proposed in the government's plans to tackle obesity.[30 64] Digital food delivery services were disproportionately advertised through digital channels, meaning that exposure to such advertising can be readily accessed from any location. Exposure to fast-food advertising was the most commonly reported of the product categories studied and no sociodemographic variation was observed, which may indicate the pervasiveness of fast-food advertising in particular.

## CONCLUSIONS

Exposure to advertising of less healthy foods and drinks was highly prevalent, with adults in lower socioeconomic groups and younger adults more likely to report exposure. Though these groups may have more objective exposure to advertising, this may also be partly due to increased receptiveness to, or recall of, advertising. Future studies should explore associations using objective measures to confirm our findings. Our findings suggest that broad advertising restrictions are likely needed to reduce exposure to HFSS advertising in adults and may help reduce sociodemographic differences in exposure to less healthy product advertising. Research evaluating the impacts of such policies should be a priority. Longitudinal studies of advertising exposure and consumption of less healthy foods and drinks would help determine whether the relationship is causal. If so, interventions should be designed to harness the potential for advertising restrictions to reduce advertising exposure in the groups that are most exposed.

**Author affiliations**
[1]Population Health Innovation Lab, Department of Public Health, Environments & Society, London School of Hygiene & Tropical Medicine, London, UK
[2]Department of Health Services Research & Policy, London School of Hygiene & Tropical Medicine, London, UK
[3]Centre for Diet & Activity Research, University of Cambridge, Cambridge, UK
[4]Department of Psychology, University of Liverpool, Liverpool, UK
[5]Population Health Sciences, University of Bristol, Bristol, UK
[6]National Institute for Health Research Applied Research Collaboration West, Bristol, UK
[7]Department of Public Health, Environments & Society, London School of Hygiene & Tropical Medicine, London, UK
[8]Centre for Public Health Research, Teesside University, Middlesbrough, UK
[9]Centre for Translational Research in Public Health (Fuse), Newcastle upon Tyne, UK
[10]Centre for Research in Public Health and Community Care, University of Hertfordshire, Hatfield, UK

**Contributors** SC, AY, JA, EJB, TB, LC, FDV, ME, VE, AAL, KL, OM, MP, CT and MW developed the research questions. SC, JA, EJB, TB, LC, FDV, ME, VE, AAL, KL, OM, MP, CT and MW contributed to the design of the survey. AY and SC designed the analysis of the survey data. AY led on the analysis with support from SC. AY led on the writing of the manuscript with support from SC. All authors critically reviewed and edited the manuscript. All authors read and approved the final manuscript.

**Funding** The National Institute for Health Research (NIHR) School for Public Health Research is a partnership between the Universities of Sheffield; Bristol; Cambridge; Imperial; and University College London; London School of Hygiene & Tropical Medicine (LSHTM); LiLaC–a collaboration between the Universities of Liverpool and Lancaster; and Fuse–Centre for Translational Research in Public Health, a collaboration between Newcastle, Durham, Northumbria, Sunderland and Teesside Universities. This study was funded by the NIHR School for Public Health Research (SPHR) (grant reference number: PD-SPH-2015). SC was also funded by Health Data Research UK (HDR-UK). HDR-UK is an initiative funded by the UK Research and Innovation, Department of Health and Social Care (England) and the devolved administrations, and leading medical research charities. JA, MW and TB were supported by the MRC Epidemiology Unit, University of Cambridge (grant number: MC/UU/12015/6) and Centre for Diet and Activity Research (CEDAR),

a UK Clinical Research Collaboration (UKCRC) Public Health Research Centre of Excellence. Funding for CEDAR from the British Heart Foundation, Cancer Research UK, Economic and Social Research Council, Medical Research Council, the National Institute for Health Research (grant numbers ES/G007462/1 and MR/K023187/1) and the Wellcome Trust (grant number: 087636/Z/08/Z), under the auspices of the UK Clinical Research Collaboration, is gratefully acknowledged. CT was funded by the NIHR Applied Research Collaboration East of England. AAL is a member of Fuse–Centre for Translational Research in Public Health (www.fuse.ac.uk). Fuse is a Public Health Research Centre of Excellence funded by the five North East Universities of Durham, Newcastle, Northumbria, Sunderland and Teesside. FDV was partly funded by NIHR Applied Research Collaboration West (NIHR ARC West) at University Hospitals Bristol NHS Foundation Trust.

**Disclaimer** The views expressed are those of the authors and do not necessarily represent those of any of the named funders. The funders had no role in the design of the study, or collection, analysis and interpretation of the data, or in the decision to publish, or in writing the manuscript.

**Competing interests** None declared.

**Patient consent for publication** Not required.

**Ethics approval** Ethical approval for this study was granted by the London School of Hygiene & Tropical Medicine Research Ethics Committee (application number: 16297/RR/11721). Written informed consent was obtained from all participants.

**Provenance and peer review** Not commissioned; externally peer reviewed.

**Data availability statement** Data may be obtained from a third party and are not publicly available. Data were collected via an online survey linked to social and demographic data from Kantar Worldpanel Plus. The terms of our data agreement with Kantar mean that we cannot share linked social and demographic data; however, access to the online survey data for the purposes of replication only may be possible. Please contact the study principal investigator (SC) (steven.cummins@ lshtm.ac.uk).

**ORCID iDs**
Amy Yau http://orcid.org/0000-0001-8889-523X
Jean Adams http://orcid.org/0000-0002-5733-7830
Laura Cornelsen http://orcid.org/0000-0003-3769-8740

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
