## [Reviewer comments · BMJ Open]

ARTICLE DETAILS

TITLE (PROVISIONAL)	Sociodemographic differences in self-reported exposure to high fat, salt and sugar food and drink advertising: a cross-sectional analysis of 2019 UK panel data
AUTHORS	Yau, Amy; Adams, J; Boyland, Emma; Burgoine, Thomas; Cornelsen, Laura; De Vocht, Frank; Egan, Matt; Er, Vanessa; Lake, Amelia; Lock, Karen; Mytton, Oliver; Petticrew, Mark; Thompson, Claire; White, Martin; Cummins, Steven

VERSION 1 – REVIEW

REVIEWER	Miranda Pallan University of Birmingham, Public Health, Epidemiology and Biostatistics
REVIEW RETURNED	29-Jan-2021

GENERAL COMMENTS	Thank you for the opportunity to review this paper, it is extremely well written, nicely presented and reports an interesting cross-sectional study. I think the authors have presented a very thoughtful interpretation of the findings in the discussion. I have very few recommendations for improvement of this paper, however I have listed a few minor suggestions to address in the discussion and conclusions: 1) Limitations – I think a fuller discussion of the limitations of this study would be helpful. For example, the BMI data are self-reported and so this limits meaningful conclusions from the BMI analyses. Only one measure of socioeconomic position was used and the majority of the sample were in the middle group. How may use of this measure as a measure of SEP influenced the findings? Ethnicity is not at all mentioned in the paper. It is plausible that ethnicity/cultural differences could influence perceived exposure to advertising, and there are complex links between ethnicity and socioeconomic disadvantage. Therefore the lack of ethnicity data could be commented on as a limitation. 2) Exposure to advertising in the last 7 days – in the discussion the authors comment that there was high exposure to advertising with 87.4% reporting that they had been exposed to advertising of less healthy foods in the last 7 days. To me this seems quite low. Given the UK environment and the breadth of advertising types that were measured, I am surprised that 12% did not report any exposure to advertising in the past week. Obviously this may link to recall and receptiveness, but possibly merits some further discussion. 3) Conclusions –it would be nice to expand a bit on the key findings of this hypothesis generating study in the conclusions section. For example, outlining that the difference in self reported exposure in
---

	lower SE groups may be due to actual exposure or differences in recall and receptiveness, and that this, and the link between advertising and consumption needs to be further explored.
REVIEWER	Mireia Montaña Blasco Universitat Oberta de Catalunya
REVIEW RETURNED	10-Feb-2021
GENERAL COMMENTS	This study explores the sociodemographic differences in self-reported exposure to less healthy food and drink advertising. This could be valuable research in light of providing a framework for food marketing regulation, especially on digital platforms. This article scientifically sounds, it is well structured, objectives are well established, the methodology is clear and statistical analysis is appropriate and fully described, and results and discussion are also complete and well thought out. It also brings a high degree of novelty to the field of study. I only have few concerns that I think could improve the paper. Firstly, in the Introduction section, some information on the audience profile of the different media should be included, to later compare if they match with the results obtained. It would also be interesting to have the advertising spending of each category studied in the different media. This way we would get an idea of the real advertising pressure that has been carried out. Secondly, because the field study was carried out during an atypical year due to the pandemic, the consumption of media in this period could have been greatly altered compared to other periods. I think this aspect should be taken into account as a limitation of the study. Finally, I think the Conclusions could go a little further in reflection, right now they are quite superficial. Despite my suggestions, I think it is a very high-quality article. I look forward to reading the final version of the manuscript.

VERSION 1 – AUTHOR RESPONSE

Reviewer: 1

Dr. Miranda Pallan, University of Birmingham

Comments to the Author:

Thank you for the opportunity to review this paper, it is extremely well written, nicely presented and reports an interesting cross-sectional study. I think the authors have presented a very thoughtful interpretation of the findings in the discussion. I have very few recommendations for improvement of this paper, however I have listed a few minor suggestions to address in the discussion and conclusions:

1) Limitations – I think a fuller discussion of the limitations of this study would be helpful. For example, the BMI data are self-reported and so this limits meaningful conclusions from the BMI analyses. Only one measure of socioeconomic position was used and the majority of the sample were in the middle group. How may use of this measure as a measure of SEP influenced the findings? Ethnicity is not at all mentioned in the paper. It is plausible that ethnicity/cultural differences could influence perceived exposure to advertising, and there are complex links between ethnicity and socioeconomic disadvantage. Therefore the lack of ethnicity data could be commented on as a limitation.

Thank you for these suggestions. We have now included the measurements of BMI and SEP used in our discussion of the limitations (lines 279-285). We also mention the lack of ethnicity data as a limitation (lines 286-287).

2) Exposure to advertising in the last 7 days – in the discussion the authors comment that there was high exposure to advertising with 87.4% reporting that they had been exposed to advertising of less healthy foods in the last 7 days. To me this seems quite low. Given the UK environment and the breadth of advertising types that were measured, I am surprised that 12% did not report any exposure to advertising in the past week. Obviously this may link to recall and receptiveness, but possibly merits some further discussion.

Thank you for your comment. We agree that self-reported exposure is likely to be an underestimation of actual exposure. We have now mentioned this in our Discussion (lines 369-373).

3) Conclusions –it would be nice to expand a bit on the key findings of this hypothesis generating study in the conclusions section. For example, outlining that the difference in self reported exposure in lower SE groups may be due to actual exposure or differences in recall and receptiveness, and that this, and the link between advertising and consumption needs to be further explored.

Thank you for this suggestion. We have now expanded our Conclusions and highlight some of the key areas that require further research (lines 386-395).

Reviewer: 2

Dr. Mireia Montaña Blasco, Universitat Oberta de Catalunya

Comments to the Author:

This study explores the sociodemographic differences in self-reported exposure to less healthy food and drink advertising. This could be valuable research in light of providing a framework for food marketing regulation, especially on digital platforms.

This article scientifically sounds, it is well structured, objectives are well established, the methodology is clear and statistical analysis is appropriate and fully described, and results and discussion are also complete and well thought out. It also brings a high degree of novelty to the field of study.

I only have few concerns that I think could improve the paper. Firstly, in the Introduction section, some information on the audience profile of the different media should be included, to later compare if they match with the results obtained. It would also be interesting to have the advertising spending of each category studied in the different media. This way we would get an idea of the real advertising pressure that has been carried out.

Thank you for this suggestion. We have now included some information about the audience profile and spend for different advertising media in the Introduction (lines 33-44).

Secondly, because the field study was carried out during an atypical year due to the pandemic, the consumption of media in this period could have been greatly altered compared to other periods. I think this aspect should be taken into account as a limitation of the study.

The survey was completed in February 2019, so the responses were not affected by the pandemic. We have now added 2019 to our title to make this clear from the beginning.

Finally, I think the Conclusions could go a little further in reflection, right now they are quite superficial.

We have now expanded the Conclusions to include some reflection on the implications of our study and future work that could be undertaken (lines 386-395).

Despite my suggestions, I think it is a very high-quality article. I look forward to reading the final version of the manuscript.

VERSION 2 – REVIEW

REVIEWER	Miranda Pallan University of Birmingham, Public Health, Epidemiology and Biostatistics
REVIEW RETURNED	22-Mar-2021

GENERAL COMMENTS	The authors have comprehensively addressed the issues I raised in my review of the original paper. This is a high quality paper which will be of interest to public health policy makers and researchers and I have no further suggestions for improvement.
---